# Laboratory Experiments and Numerical Simulation on Dynamic Response of Island Reclamation Coral Sand under Aircraft Load

**DOI:** 10.3390/ma16093465

**Published:** 2023-04-29

**Authors:** Jianxiu Wang, Ansheng Cao, Dongsheng Song, Bo Feng, Huboqiang Li, Yanxia Long, Zhenhua Ye

**Affiliations:** 1College of Civil Engineering, Tongji University, Shanghai 200092, China; 2110410@tongji.edu.cn (A.C.); dj12sds@tjad.cn (D.S.); 2210056@tongji.edu.cn (H.L.); 2010328@tongji.edu.cn (Y.L.); 2Key Laboratory of Geotechnical and Underground Engineering of Ministry of Education, Tongji University, Shanghai 200092, China; 3Greenland Infrastructure Construction Group Co., Ltd., Shanghai 200011, China; bo.feng@foxmail.com

**Keywords:** coral sand, aircraft load, dynamic triaxial compression test, numerical simulation, dynamic response

## Abstract

The construction of island airports on coral reefs inevitably encounters the impact load of aircraft takeoff and landing. However, previous studies have not presented a detailed description of the dynamic response of the coral sand beneath the runways of island reclamation airports under aircraft load. In the current study, the coral sand of Mischief Reef Airport in the Nansha Islands, China, was selected as the background. The pore water pressure and strain characteristics of reshaped coral sand under aircraft loads with different dynamic stress amplitudes and vibration frequencies were studied using dynamic triaxial tests. Particle discrete element software was employed to study the deformation characteristics of coral sand with different particle sizes and porosities under aircraft loads. Results show that when the dynamic stress amplitude and vibration frequency were small, the pore water pressure and strain of the coral sand samples gradually increased with the number of load cycles, and the growth rate became increasingly small. When the dynamic stress amplitude and vibration frequency were large, the axial strain of the coral sand samples increased with the vibration frequency, and the growth rate exhibited an increasing trend. The deformation of the coral sand samples increased with porosity under aircraft loading. The larger the variation range of the coral sand particle size was, the larger the coral sand deformation caused by aircraft takeoff and landing load was. These results can provide a reference for the treatment and repair of the airstrip foundation of island airports.

## 1. Introduction

The coral reefs of the Nansha Islands are the most valuable land resource in the South China Sea. The reefs are the foothold and forward base for the exploitation of marine resources. Coral reefs are the preferred construction sites for island airports. Compared with general airport runway pavements, the runways of island airports often need to be reclaimed due to the limitations of island sites. A coral reef island reclamation airport runway is characterized by high water content, loose structure, a large void ratio, high compressibility, and low natural strength [1,2]. This runway can easily produce a large or uneven settlement under large loads, short-acting time, and low frequency of aircraft takeoff and landing. Therefore, the dynamic response of coral sand in island reclamation airport runways under aircraft load must be understood to construct airports on coral reefs.

Coral sand is mainly composed of coral debris and other marine biological debris, and the calcium carbonate content reaches 90%. The unique properties of coral sand are due to its special material composition, structure, and development environment [3,4,5]. Scholars have conducted studies on the characteristics of coral sand. Some have studied the microscopic structure of coral sand through electron microscopy, CT scanning, and mercury intrusion tests [6,7,8]. They found that coral sand particles are mainly composed of coral reef debris with different particle sizes and original biological textures. The roundness of coral sand is different from that of inland sand with an extremely irregular shape and low hardness, causing the coral sand particles to easily break at the edges and corners. Coral sand is more prone to particle breakage than ordinary sand, and continuous particle breakage causes the grading of coral sand to constantly change, thus affecting its properties, such as shear expansion, compressibility, and critical state, and ultimately threatening project safety [9,10,11,12]. Sharma et al. [13] studied the static characteristics of coral sand by using triaxial compression equipment, and their results showed that the dynamic strength of calcareous sand is related to its relative density and effective confining pressure, and it is particularly sensitive to confining pressure. Wang et al. [14] found that the hydraulic conductivity of coral sand increases linearly with particle size. Rui et al. [15] reported that the compression modulus of a sample changes with the increase in coral sand’s rod-like or lamellar particles, but the irregular property of coral sand can improve the sample’s resistance to soil liquefaction. Aissa et al. [16] studied the fatigue characteristics of coral aggregate concrete under static compaction and revealed the good adhesion between the calcareous tuff and cement matrix. Sandoval et al. [17] compared the normalized pore pressure development curves of calcrete and quartz sand under the same confining pressure and relative density through an equal consolidation triaxial compression test and found that calcrete sand presents a large pore pressure fluctuation, and its pore pressure development curve is convex at the top, whereas that of quartz sand is concave at the bottom. Shahnazari et al. [18] systematically studied the evolution law of coral sand particle breakage during triaxial shear from the perspectives of particle size, compactness, confining pressure, shear strain, drainage conditions, and energy and found that input energy plays an important role in particle breakage behavior. Coop et al. [19] investigated the particle breakage degree of coral sand under different shear strains through a ring shear test and discovered that coral sand no longer breaks after reaching a certain shear strain; the initial gradation and normal stress of the sample are the main factors that determine the final gradation. Wang et al. [20] conducted a direct shear test on calcareous sand in the South China Sea and found that the cohesion of calcareous sand is much higher than that of quartz sand and increases considerably with the average particle size. Rezvani et al. [21] compared the triaxial shear behaviors of two kinds of coral sand with different particle sizes under the same conditions and found significant differences in their stress–strain curves. Notably, the abovementioned research only studied the mechanical properties of coral sand from a static or quasistatic perspective.

The dynamic characteristics of coral sand must be explored to ensure that buildings with a coral sand foundation can withstand strong dynamic loads, such as earthquakes, strong impacts, accidents, or man-made explosions. Dong et al. [22] conducted an SHPB impact experiment and found that coral sand exerts an obvious strain rate effect, and the strain rate sensitivity is closely related to the inner pores and the intergranular friction. Coral sand behaves differently from ordinary terrestrial sand under dynamic loads, such as earthquakes, tsunamis, and waves. Many scholars have studied the characteristics of coral sand under the dynamic cyclic load of earthquakes, tsunamis, and waves by using a triaxial apparatus [23,24]. Airey et al. [25] examined the cycling performance of coral calcareous sand in Northwest Australia and found that the frequency of loading affects the pore pressure response measured at the ends of samples. Sadrekarimi et al. [26] conducted a triaxial drainage cyclic loading test on coral sand and discovered that the particle breakage degree increases with the number of cycles. Wang et al. [27] performed multiple consolidation and drainage triaxial shear tests to explore the effects of particle size and confining pressure on the shear characteristics of coral sand.

The structure of coral sand is characterized by nonuniformity, discontinuity, and anisotropy. The conventional finite element method has certain limitations when used to study coral sand’s mechanical properties, and the mechanical behavior of coral sand is difficult to determine from the microscopic perspective. The discrete element method (DEM) proposed by Cundall can determine the crushing characteristics of granular materials [28]. The particle flow code (PFC) developed based on DEM has become an important simulation method and has been widely used in the numerical simulation of sand particles. Xu et al. [29] studied the impact of particle size distribution on the impact resistance of cemented coral sand by using DEM. Their results showed that the impact resistance of cemented coral sand increases sharply at first then gradually decreases with increasing average particle size and uniformity coefficient. Alshibli et al. [30] applied DEM to simulate a direct shear test on granular materials from the perspective of microscopic mechanics, with focus on the influence of shear sample size on the macroscopic properties of materials. Chen et al. [31] used PFC to conduct a numerical simulation of the triaxial compression test on sand, and the results showed that the microscopic parameters of sand markedly affect the macroscopic response of sand. Zhang et al. [32] used DEM to study the influence of sampling factors on the triaxial shear performance of cemented sand and found that peak strength and pre-peak stiffness are enhanced by decreasing the initial void ratio. Shamy et al. [33] used the fluid–particle coupling method to study the liquefaction of saturated sand under dynamic loads and provided valuable information on a number of salient microscale response mechanisms. Jiang et al. [34] conducted biaxial tests on structural loess with different water contents with the help of PFC to explore the collapsibility characteristics of loess and loess-like materials under complex stress paths.

Research on the mechanical properties of coral sand under static and dynamic loads has achieved fruitful results, but research on the mechanical characteristics of coral sand under dynamic cyclic loads is limited, and research on the dynamic response of coral sand under dynamic aircraft loads is even rarer. Additionally, the dynamic characteristics of coral sand are affected by the loading mode and coral sand’s own microstructure, particle size, porosity, and other factors. To fully explore the dynamic response of coral sand under cyclic dynamic loading, this study combined the dynamic triaxial compression test and particle discrete element analysis to study the dynamic response of coral sand under different dynamic stress amplitudes, vibration frequencies, particle sizes, and porosities.

## 2. Laboratory Experiment

### 2.1. Coral Sand

The coral sand in this study was obtained from Mischief Reef in the South China Sea. Mischief Reef is located in the middle east of Nansha Islands in the South China Sea. The geographical location of the reef is 9°54′00″ north latitude and 115°32′00″ east longitude, as shown in Figure 1a. As indicated in Figure 1b, Mischief Reef is a typical oval coral atoll that is approximately 9 km from east to west and 6 km from north to south, with a total area of about 46 km^2^. This reef is a rare, large, natural shelter harbor. In 2015, the People’s Republic of China began the reclamation and construction of an airfield with a 55 m wide and 2660 m long runway in the west of Mischief Reef.

The coral sand obtained from Mischief Reef was analyzed in terms of particle size, and coral sand samples with different particle sizes were obtained, as shown in Figure 2. During the mechanical test on coral sand, only particles less than 2 mm, which accounted for the majority of the particles, were tested to control the test conditions and facilitate the analysis of results. Figure 3a,b show the scanning electron microscopy (SEM) images of coral sand with particle sizes less than 0.075 and 2 mm, respectively. According to the SEM images, the clay and sand particles presented flaky, block, and dendritic shapes. The flaky grain structure was dense, with few or no internal pores and a small content in the sand sample, and its shape was regular. The dendritic particles retained the protozoa form with primary and corrosion pores, and their shape was regular. The particles were extremely complex in shape, contained internal pores, and had a large content in the sand sample.

According to the standard for soil testing (GB/T 50123-2019) [35], the maximum particle size should not exceed one-tenth of the diameter of the sample. In this experiment, the sample size was 39.1 mm × 80 mm, so the sample particle diameter was 2 mm at the most. In the experiment, particles larger than 2 mm were removed by a removal method. The particle size distribution of coral sand after removing particles larger than 2 mm is shown in Table 1. Consolidated undrained triaxial shear tests were performed on the samples at confining pressures of 30, 70, 100, and 200 kPa. The stress–strain relationship curves of coral sand with different confining pressures are shown in Figure 4. The stress–strain curves presented a linear change at the initial stage, and the strain increased with increasing stress. In the subsequent shear process, the shear stress decreased gradually with increasing strain, and the strain decreased slightly after reaching the peak value. Mohr stress circles with different confining pressures obtained based on the Mohr–Coulomb strength formula, and test results are shown in Figure 5. The shear strength envelope was derived using the common tangent of the Mohr stress circle. The internal friction angle (*φ*) was 40.174°, and the cohesion (*c*) was 14.258 kPa.

### 2.2. Sample Preparation

Original samples of coral sand are difficult to obtain, so a dynamic triaxial compression test was conducted on saturated soil by using reshaped samples with dry density as the main control index. Unlike ordinary quartz sand, coral sand is mainly composed of calcium carbonate and easily broken, so a new sample preparation equipment was designed to prepare samples directly on the dynamic triaxial equipment and saturate them, as shown in Figure 6. To eliminate the influence of different loading methods as much as possible, the layered wet loading method was employed to prepare the samples. During sample preparation, the permeable stone, rubber membrane, and split round membrane were successively placed on the base of the pressure chamber. At a coral sand dry density of 1.41 g/cm^3^ and sample volume of 90.0 cm^3^, the required sand sample mass was 127.0 g. The sand was boiled in water and divided into three parts after cooling. The mold was filled with pure water to 1/3 of the sample height, and the rubber film was filled with the first sand sample to the required height of the layer. The same steps were performed to fill the second and third layers with sand until the film was filled. External vacuuming equipment was installed to vacuum the sample. After approximately an hour of vacuum extraction, the sample was fully saturated; the vacuum instrument was removed; the sample top cap was installed; and the external pressure chamber for the experiment was installed.

### 2.3. Aircraft Landing Load

Aircraft load is cyclic [36]. The aircraft load in a single cycle is represented by a half-sine load, as shown in Figure 7. The coral sand layer 4–6 m below the airport runway was experimentally studied during aircraft takeoff and landing. The coral sand on the airport runway at this depth was easily crushed under the aircraft loads, resulting in severe liquefaction and lateral slippage, which led to dynamic instability of the airport runway. The dynamic stress amplitude of coral sand 4–6 m below the airport runway during aircraft takeoff and landing (*σ*_d_) was 70–150 kPa, and the vibration frequency was 0.5–2 s.

### 2.4. Experimental Scheme

The experiment aimed to study the dynamic characteristics of saturated coral sand subjected to an aircraft cyclic vibration load in an undrained state. The dynamic characteristics of soil are affected by many conditions, including the dynamic stress amplitude, confining pressure, consolidation ratio, vibration frequency, number of cycles, and saturation. In accordance with the experimental purpose and requirements, the amplitude and frequency of aircraft cyclic dynamic stress were set as variables to study the dynamic response of the coral sand runway of the island airport. Two working conditions were set up in the experiment, as shown in Table 2. In Working Condition 1, the consolidation confining pressure, cyclic load frequency, coral sand gradation, and compactness were unchanged, whereas the cyclic dynamic stress amplitude was changed. In Working Condition 2, the cyclic load amplitude, consolidation confining pressure, coral sand gradation, and compactness were unchanged, whereas the cyclic load frequency was changed. The object of the experiment was coral sand 4–6 m below the airport runway, and the confining pressure of coral sand at this depth was approximately 100 kPa. Therefore, the coral sand confining pressure in Conditions 1 and 2 was set to 100 kPa.

### 2.5. Experimental Procedures

A cyclic loading experiment was performed on the coral sand samples in accordance with the standard soil testing method (GB/T 50123-2019) [35] by using the dynamic triaxial test system shown in Figure 8. First, 20 kPa of confining pressure was applied, and at a certain water head height, airless water flowed through the sample from the bottom to the top for 30 min. The Sation module was used for back pressure saturation for 120 min. After saturation, Skempton B was detected. When the Skempton B value exceeded 0.95, the sample was considered to be saturated and to have met the saturation requirements. Otherwise, back pressure was still applied to ensure that the soil sample met the requirements. Second, 100 kPa of confining pressure was applied, and the sample was consolidated using the advanced triaxial test module. The consolidation time was approximately 24 h. The consolidation was completed when the axial strain was less than 0.05%. The dynamic triaxial test module was used for aircraft load loading; that is, dynamic stress was applied in the axial direction. Test data were automatically collected by a computer, and the time history data of the consolidation process, the vibration pore pressure, the dynamic load, and the dynamic strain were recorded. The strain failure criterion was selected as the liquefaction standard of coral sand [37]; that is, the test ended when the axial strain reached 5%.

### 2.6. Results and Discussion

#### 2.6.1. Influence of Dynamic Stress Amplitude on the Dynamic Response of Coral Sand

The dynamic triaxial test results of coral sand with dynamic stress amplitudes (*σ*_d_) of 70, 80, 90, and 145 kPa in Working Condition 1 were analyzed to study the influence of the dynamic stress amplitude on the dynamic response of coral sand. Figure 9 and Figure 10 show the pore water pressure and strain time history curves of coral sand when the dynamic stress amplitude was 70, 80, 90, and 145 kPa.

When the other test conditions were the same and the dynamic stress amplitude was small (*σ*_d_ = 75, 80, 100 kPa), the pore water pressure and strain gradually increased with the number of vibrations, but the growth rate gradually decreased. Finally, the pore water pressure and strain did not increase with the increasing vibration number (Figure 9 and Figure 10a–c). When a certain number of vibration times was reached, the coral sand sample was compacted to a certain degree and had enough compactness to resist the external load. Thereafter, the axial strain did not change, and the sample deformation was small and did not reach failure state. When the dynamic stress amplitude was large (*σ*_d_ = 145 kPa), the axial strain increased with the vibration frequency, and the growth rate became increasingly large. The strain soon increased to the maximum strain of 5%, reaching the failure (liquefaction) state.

The maximum pore water pressure and strain of the sample when the dynamic stress amplitude was 70, 80, and 90 kPa during the test are shown in Table 3. The relationships between dynamic stress amplitude, maximum pore pressure, and maximum strain are shown in Figure 11 and Figure 12. Under cyclic loading, the maximum pore water pressure and strain of coral sand increased linearly with the dynamic stress amplitude. This result is consistent with Peng’s experimental results for transparent sand [38]. The reason for this phenomenon was that the larger the dynamic stress amplitude was, the more sufficient the vibration between the sample particles was, and the easier the relative sliding and rotation between the sample particles were. Moreover, the axial strain increased due to the rearrangement of the particles.

#### 2.6.2. Influence of Cyclic Load Vibration Frequency on the Dynamic Response of Coral Sand

The dynamic triaxial test results on coral sand under different cyclic load vibration frequencies (0.5, 0.8, 1, 1.2, and 2 Hz) were analyzed to study the influence of vibration frequency on the dynamic response of coral sand. Figure 13 and Figure 14 show the pore water pressure and axial strain time history curves of coral sand when the vibration load frequency *f* = 0.5, 0.8, 1, 1.2, and 2 Hz. The test conditions were as follows: the consolidation pressure was 100 kPa; the consolidation ratio was 1; the void ratio was 1.07; and the dynamic stress amplitude was 100 kPa.

Comparison of the pore water pressure and strain–time history curves at vibration frequency *f* = 0.5, 0.8, 1, 1.2, and 2 Hz showed that under the same consolidation pressure, consolidation ratio, void ratio, and dynamic stress amplitude, when the vibration frequency was small (*f* = 0.5, 0.8, 1, 1.2 Hz), the pore water pressure and strain gradually increased with the vibration number, but the growth rate became increasingly small and eventually stabilized. When the vibration frequency was large (*f* = 2 Hz), the axial strain increased with the vibration frequency; the growth rate was increasingly large; and the strain soon increased to the maximum of 5%, reaching the failure (liquefaction) state. When the vibration frequency was small, the sample deformation was small and did not reach the failure state. When the vibration frequency was too large, the pore water pressure and deformation of the sample increased rapidly, and the sample structure was destroyed.

The maximum pore water pressure and strain of the coral sand when vibration frequency *f* = 0.5, 0.8, 1, and 1.2 Hz are listed in Table 4. Figure 15 and Figure 16 show the relationship curves between vibration frequency and maximum pore water pressure and strain, respectively.

The maximum pore water pressure increased with the vibration frequency, and the fitting equation of frequency and maximum pore water pressure is expressed as
y = 12.851ln(x) + 58.733.(1)

The maximum strain increased with the vibration frequency and presented an obvious nonlinear relationship as follows:y = 0.9266x^2^ − 1.2161x + 1.4907.(2)

The maximum strain and pore water pressure nonlinearly increased with increasing vibration frequency. The analysis indicates that the high vibration frequency represented high loading and unloading speeds. The specimen with a high loading frequency had a long loading time, so it had a large deformation.

## 3. Numerical Simulations

DEM is an important approach to study the micromechanical properties of rock and soil granular materials. Compared with laboratory testing, DEM enhances the controllability and repeatability of the initial state and loading conditions of the sample and can more accurately analyze the influence of various microscopic parameters on the mesoscale response of the characterization volume element.

### 3.1. Modeling of the Numerical Model

On the basis of the experimental samples, a 2D model of coral sand with a height of 80 mm and a diameter of 39.1 mm was established using the Particle Flow Code in 2 Dimensions V5.0software. The establishment of the numerical model included the generation of walls and particles. The upper, lower, left, and right walls were generated based on the model size of 80 mm × 39.1 mm. At this time, the positions of the walls were fixed, and the walls constrained the particles during the creation of the model. Particles were generated in the wall area. Coral sand particles are generally nonspherical but considered to be circular and scaled up to improve computational efficiency during numerical simulations [39,40]. Therefore, in this study, the particles were summarized as spherical, and the size of the coral sand specimen was equally calculated. Many particles with a particle size less than 0.1 mm were observed (Table 1), and they could not be fully generated in the simulation. Therefore, the particles with a particle size less than 0.1 mm were simplified into the bonding force between particles. In the numerical model, the minimum and maximum particle sizes of the coral sand particles were 0.1 and 5.0 mm, respectively. The entire model had 6598 particles, as shown in Figure 17. The sample adopted isotropic consolidation, and a constant confining pressure of 100 kPa was applied to the sample by continuously adjusting the radial velocity toward the side wall. The radial velocity of the side wall was automatically controlled by a numerical servo mechanism. The axial load in the experiment was a semisine cyclic load, whose dynamic stress amplitude and loading frequency were 100 kPa and 1 Hz, respectively. The required axial semisine cyclic load was converted into velocity and applied to the lower wall through the servo control mechanism.

### 3.2. Numerical Model Parameters

The linear stiffness model, which can simulate the stiffness characteristics of sand, was used to describe the contact between the coral sand particles. The microscopic parameters in the linear stiffness model included normal stiffness *k_n_*, tangential stiffness *k_s_*, friction coefficient *μ*, and porosity *n*. The microscopic parameters were the particle contact parameters inside the sample; these parameters can reflect the mechanical properties between particles. The macroscopic mechanical parameters represented the macroscopic mechanical properties when the sample was regarded as a whole. The macroscopic mechanical parameters of the coral sand samples were obtained from laboratory compression and triaxial shear tests, as shown in Table 5. The microscopic parameters were calibrated by the macroscopic parameters.

The microscopic parameters, such as normal and tangential stiffness of the spherical particles, were initially determined. The microscopic parameters were substituted into the numerical model to conduct a laboratory compression test simulation, and the stress–strain and volumetric strain and axial strain curves were obtained. The Poisson’s ratio and elastic modulus were calculated based on the strain curve and compared with those of the macroscopic parameters. When the deviation between the two was large, the microscopic parameters were changed, and the simulation was performed again. The macroscopic mechanical parameters calculated by the numerical simulation were consistent with those obtained by laboratory tests. The micromechanical parameters of the coral sand sample are shown in Table 6.

### 3.3. Numerical Simulation Scheme

Numerical simulation was performed to study the effects of different particle sizes and porosities on the dynamic response of coral sand. When studying the influence of porosity on the dynamic response of the coral sand, the particle size was controlled between 1.8 and 4.7 mm, and the porosity was set to 0.28, 0.30, 0.32, 0.34, and 0.36 for five simulations. When studying the influence of particle size on the dynamic response of coral sand, the porosity was controlled to 0.3 to generate particles, and the particle size was controlled to 1.8–4.7, 2.0–4.5, 2.2–4.3, 2.4–4.1, 2.6–3.9, and 2.8–3.7 mm.

### 3.4. Numerical Reliability Analysis

Figure 18 shows the strain–time curves of the experimental and numerical simulations under similar conditions. The shapes of the two strain curves were similar. At the initial stage of cyclic loading, the growth rate of strain was large, and the strain growth rate decreased with the increase in the number of cycles, indicating that the numerical simulation could simulate the strain hardening characteristics of the coral sand with the cyclic load. The numerical simulation results were similar to the experimental results, indicating that the mesoscopic parameters were reasonable.

### 3.5. Numerical Simulation Results and Discussion

#### 3.5.1. Influence of Porosity on the Dynamic Response of Coral Sand

Figure 19, Figure 20 and Figure 21 show the axial displacement, radial displacement, and volume variables of coral sand with different porosities at the end of the simulation.

The axial displacement and volume variables of the coral sand samples increased considerably with porosity (Figure 17 and Figure 19). The sample with a large porosity had relatively loose characteristics. The contact among the loose particles was also less than that among the dense ones. The ability of the loose particles to resist deformation was weak. The larger the porosity was, the greater the deformation was. The radial displacement of the coral sand samples did not change considerably with increasing porosity (Figure 18). Porosity had a remarkable effect on the deformation of the coral sand under a dynamic load. The settlement of the coral sand increased with porosity.

#### 3.5.2. Influence of Particle Size on Dynamic Response of Coral Sand

The axial displacement, radial displacement, and volume variation of the coral sand with different particle sizes at the end of the simulation are shown in Figure 22, Figure 23 and Figure 24.

The axial displacement of the coral sand sample decreased with the particle size range, and the reduction rate gradually increased (Figure 22). The sample particles were uniform and within a small particle size range; they were difficult to compact under external loads. Therefore, the smaller the sample size range was, the smaller the deformation was. Radial displacement decreased with the particle size range (Figure 23). Moreover, the volume variable increased and decreased irregularly with the reduction in the particle size range (Figure 24).

## 4. Conclusions

Coral sand is used for the runways of island reclamation airports. In this study, the dynamic response of coral sand under aircraft takeoff and landing loads was investigated using experimental and numerical simulation methods. The following conclusions were derived.

(1)When the amplitude of the aircraft cyclic load was small, the pore water pressure and strain of coral sand gradually increased with the number of load cycles, and the growth rate decreased gradually until it stabilized. When the amplitude of the aircraft cyclic load was large, the strain of coral sand increased with the number of load cycles, and the growth rate continuously increased.(2)When the frequency of the aircraft cyclic load was low, the pore water pressure and strain of coral sand gradually increased with the number of load cycles, but the growth rate gradually decreased. When the vibration frequency of the aircraft cyclic load was high, the strain of coral sand increased with the number of load cycles, and the growth rate increased.(3)Under the aircraft cyclic load, the maximum pore water pressure and strain of coral sand increased linearly with the dynamic load amplitude and increased nonlinearly with the vibration frequency.(4)The porosity of the coral sand particles considerably affected the deformation of coral sand under aircraft cyclic loading, and the deformation increased with porosity.(5)The smaller the variation range of the coral sand particle size was, the smaller the coral sand settlement caused by aircraft takeoff and landing loads was.

## Figures and Tables

**Figure 1 materials-16-03465-f001:**
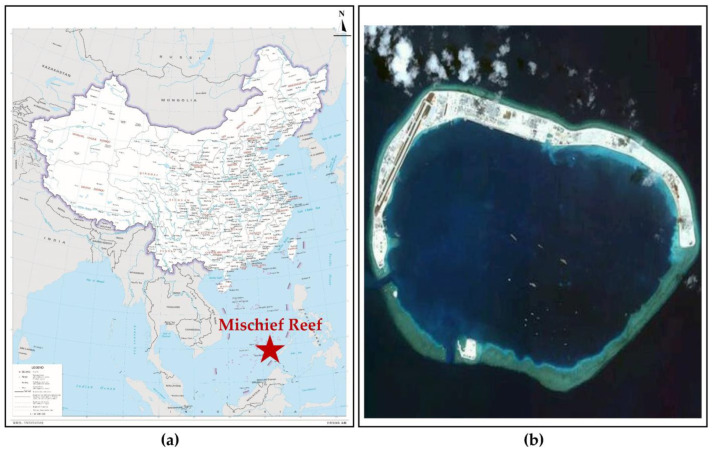
Overview of Mischief Reef: (**a**) Location of Mischief Reef; (**b**) Satellite view of Mischief Reef.

**Figure 2 materials-16-03465-f002:**
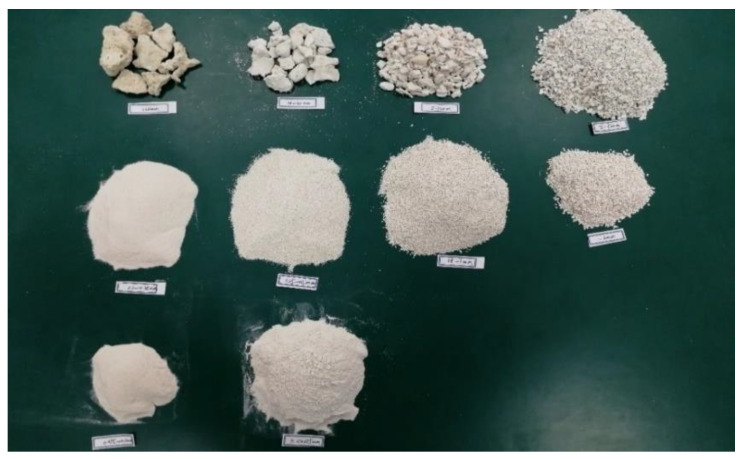
Coral sand with different particle sizes.

**Figure 3 materials-16-03465-f003:**
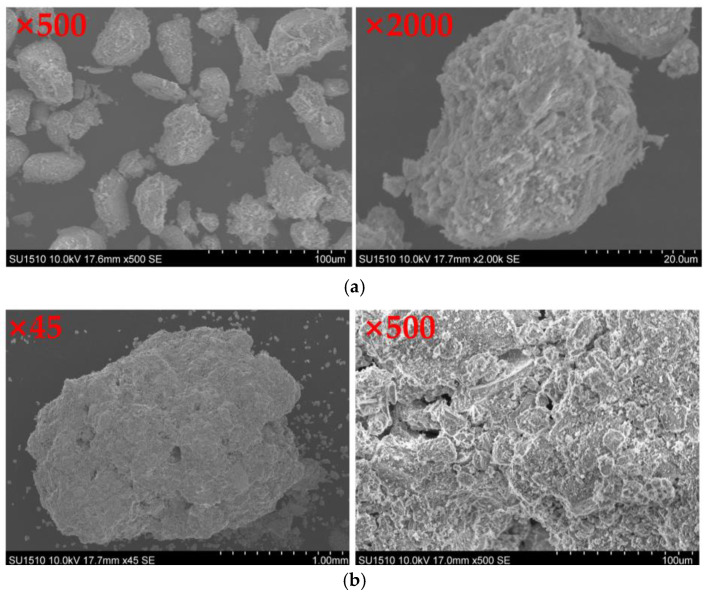
Electron microscope scanning of coral sand: (**a**) clay particles (diameter < 0.075 mm); (**b**) sand particles (diameter < 2 mm).

**Figure 4 materials-16-03465-f004:**
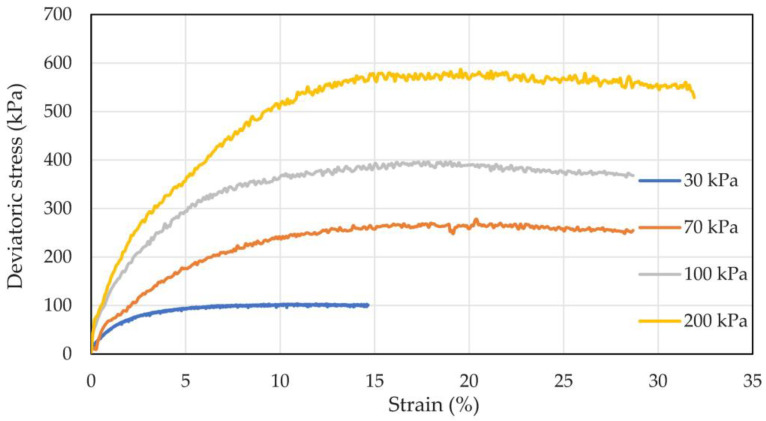
Stress–strain curve of the consolidated undrained shear test.

**Figure 5 materials-16-03465-f005:**
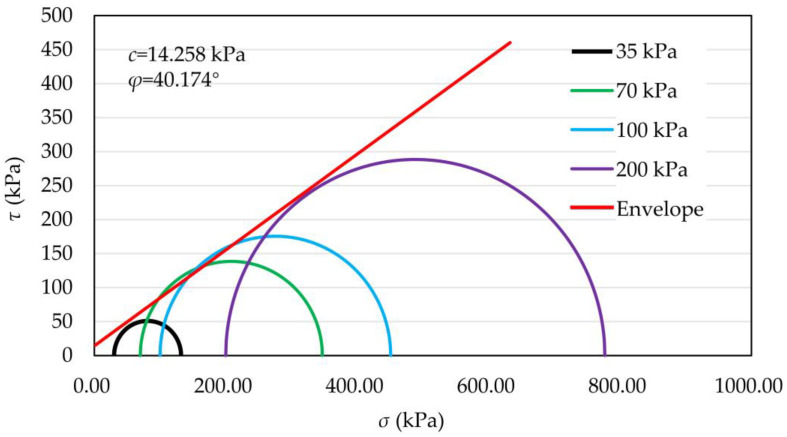
Shearing strength envelope.

**Figure 6 materials-16-03465-f006:**
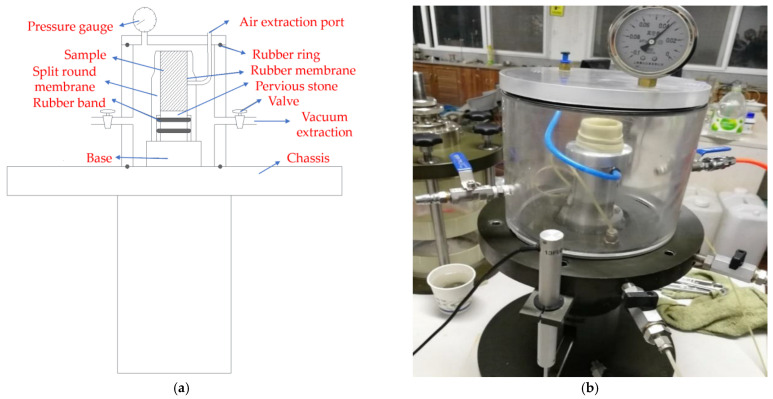
Dynamic triaxial sample preparation and vacuum pumping equipment: (**a**) design drawing; (**b**) physical drawing.

**Figure 7 materials-16-03465-f007:**
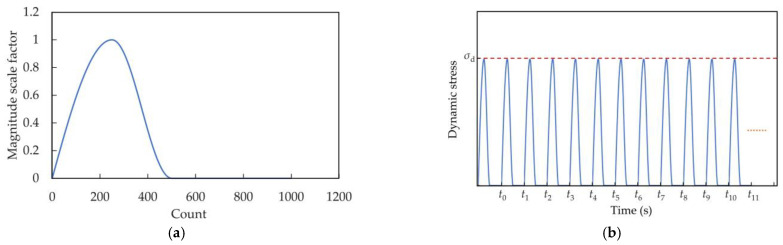
Aircraft vibration load curve: (**a**) single period; (**b**) cycle period.

**Figure 8 materials-16-03465-f008:**
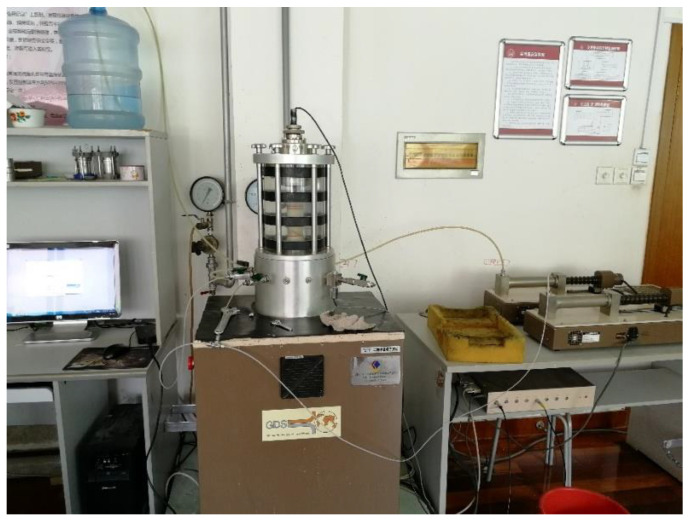
Dynamic triaxial test system.

**Figure 9 materials-16-03465-f009:**
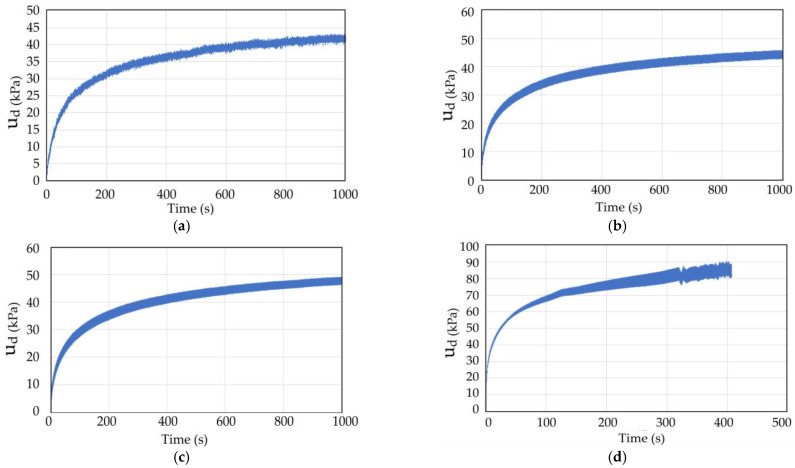
Pore water pressure time history curve of coral sand with different dynamic stress amplitude: (**a**) *σ*_d_ = 70 kPa; (**b**) *σ*_d_ = 80 kPa; (**c**) *σ*_d_ = 90 kPa; (**d**) *σ*_d_ = 145 kPa.

**Figure 10 materials-16-03465-f010:**
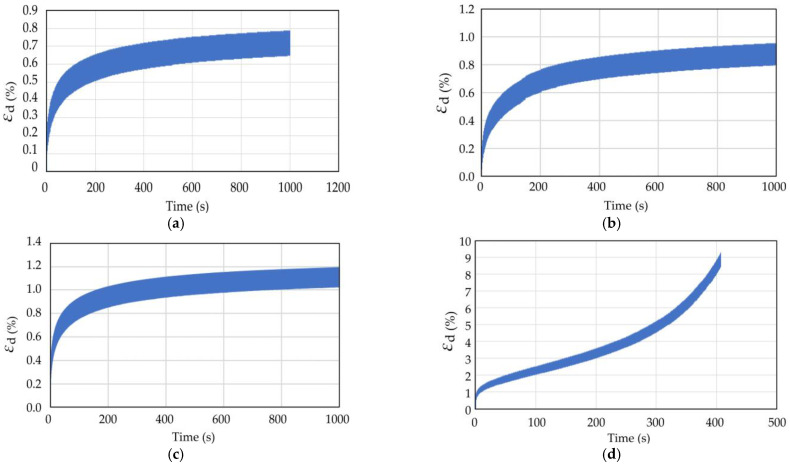
Strain time history curve of coral sand with different dynamic stress amplitude: (**a**) *σ*_d_ = 70 kPa; (**b**) *σ*_d_ = 80 kPa; (**c**) *σ*_d_ = 90 kPa; (**d**) *σ*_d_ = 145 kPa.

**Figure 11 materials-16-03465-f011:**
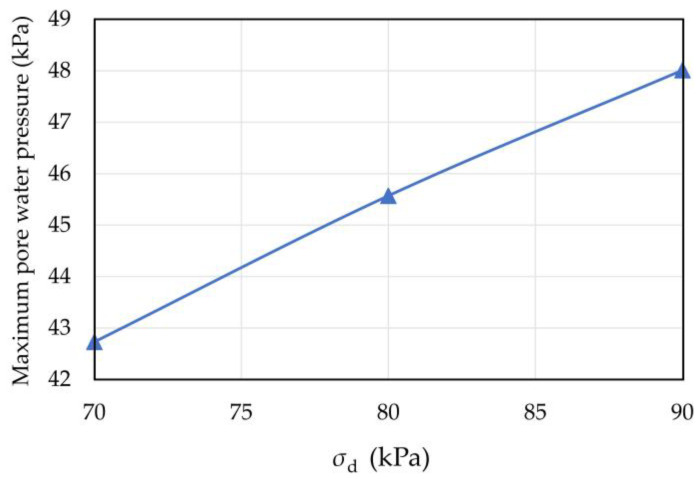
Relationship curve between dynamic stress amplitude and maximum pore water pressure of coral sand.

**Figure 12 materials-16-03465-f012:**
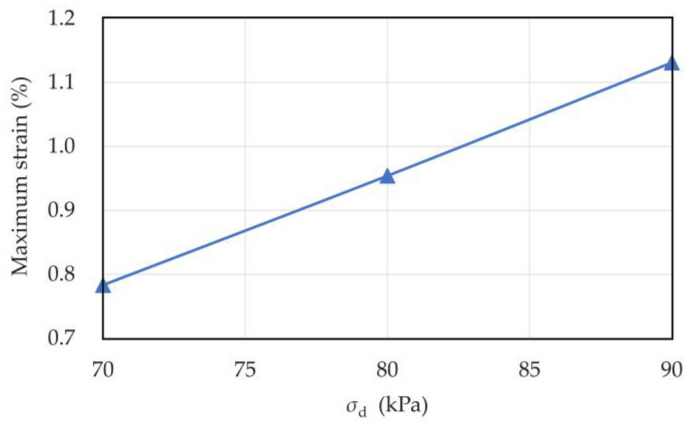
Relationship curve between dynamic stress amplitude and maximum strain of coral sand.

**Figure 13 materials-16-03465-f013:**
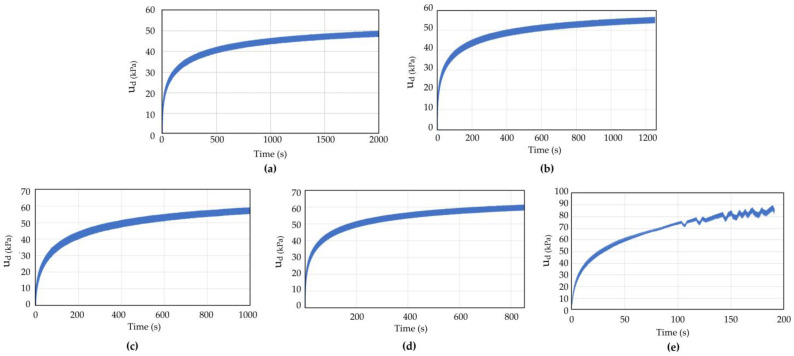
Pore water pressure time history curve of coral sand with different vibration frequencies: (**a**) *f* = 0.5 Hz; (**b**) *f* = 0.8 Hz; (**c**) *f* = 1.0 Hz; (**d**) *f* = 1.2 Hz; (**e**) *f* = 2.0 Hz.

**Figure 14 materials-16-03465-f014:**
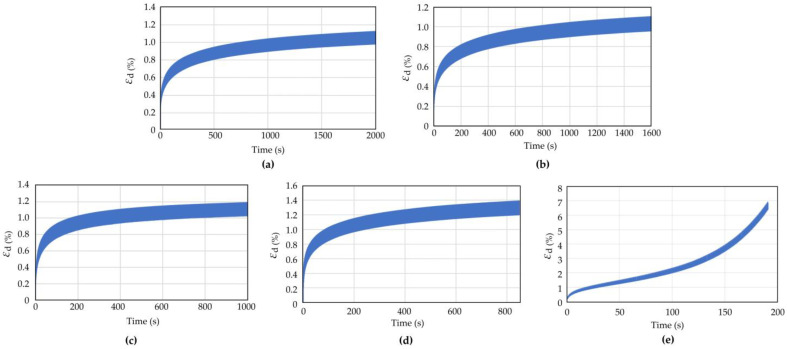
Strain time history curve of coral sand with different vibration frequencies: (**a**) *f* = 0.5 Hz; (**b**) *f* = 0.8 Hz; (**c**) *f* = 1.0 Hz; (**d**) *f* = 1.2 Hz; (**e**) *f* = 2.0 Hz.

**Figure 15 materials-16-03465-f015:**
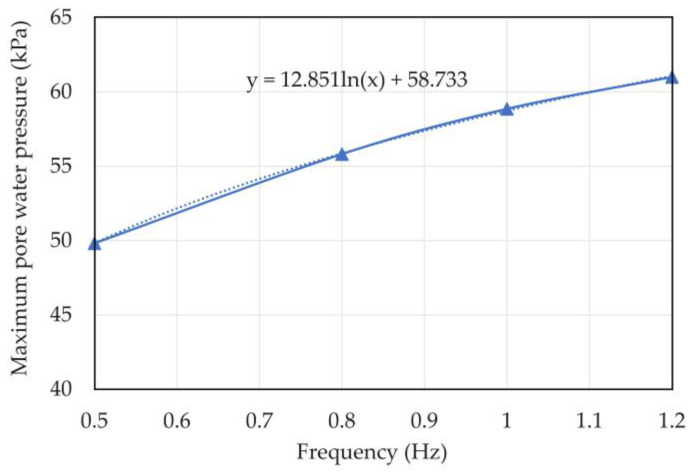
Relationship curve between vibration frequency and maximum pore water pressure of coral sand.

**Figure 16 materials-16-03465-f016:**
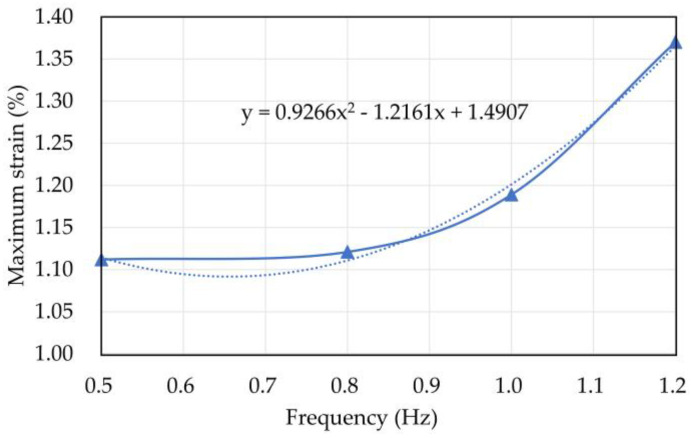
Relationship curve between vibration frequency and maximum strain of coral sand.

**Figure 17 materials-16-03465-f017:**
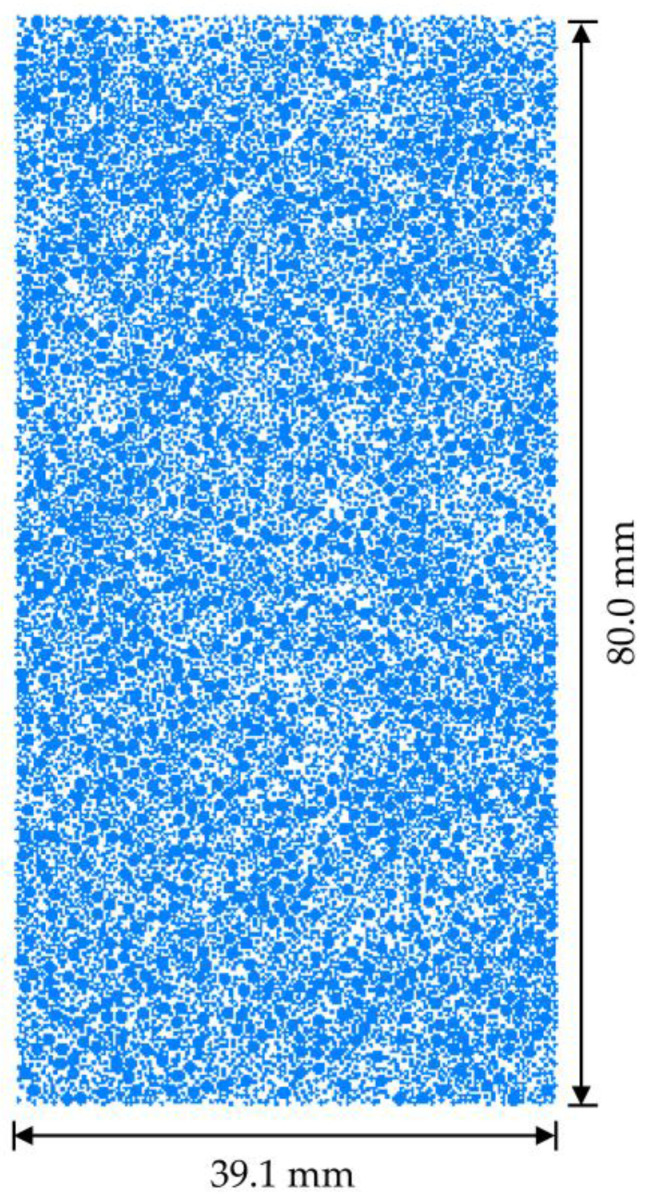
Numerical simulation sample.

**Figure 18 materials-16-03465-f018:**
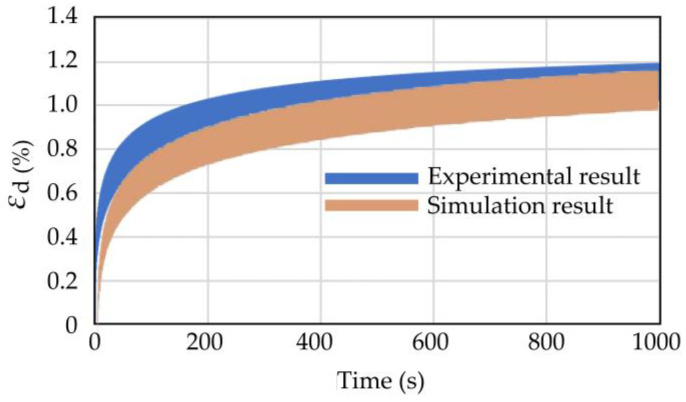
Comparison of strain time history curve between experimental and numerical simulation.

**Figure 19 materials-16-03465-f019:**
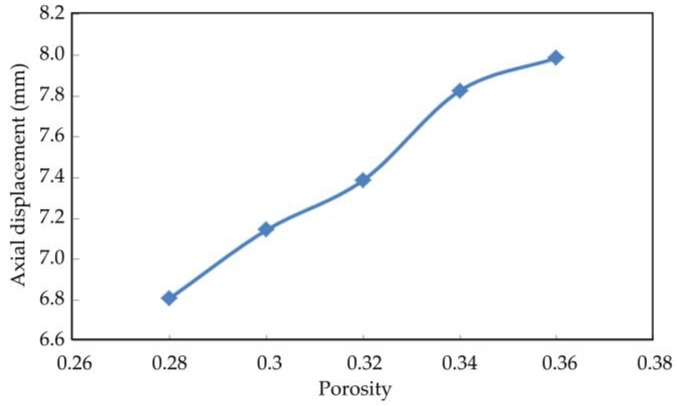
Relationship between axial displacement and porosity.

**Figure 20 materials-16-03465-f020:**
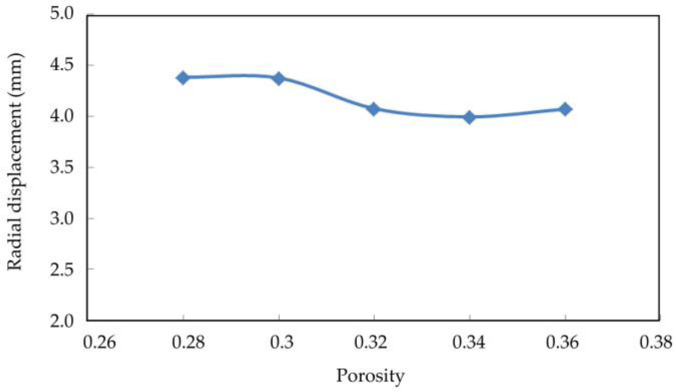
Relationship between radial displacement and porosity.

**Figure 21 materials-16-03465-f021:**
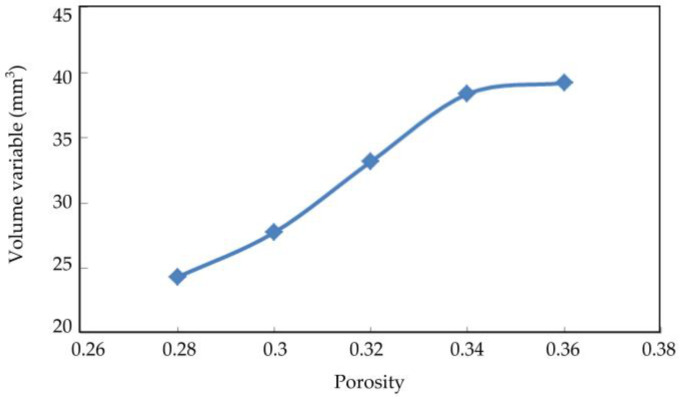
Relationship between volume variables and porosity.

**Figure 22 materials-16-03465-f022:**
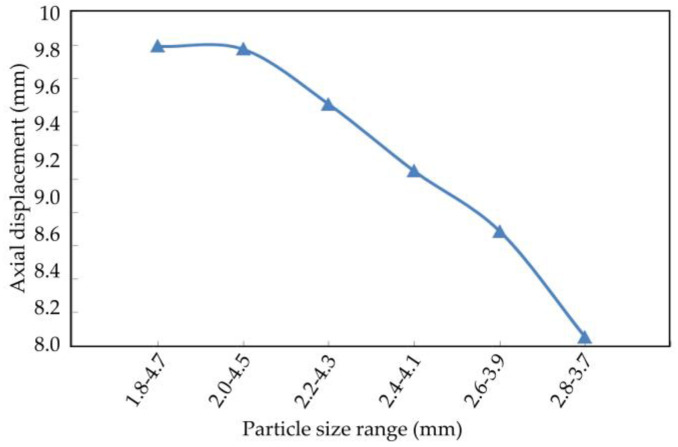
Relationship between axial displacement and particle size. 2013.

**Figure 23 materials-16-03465-f023:**
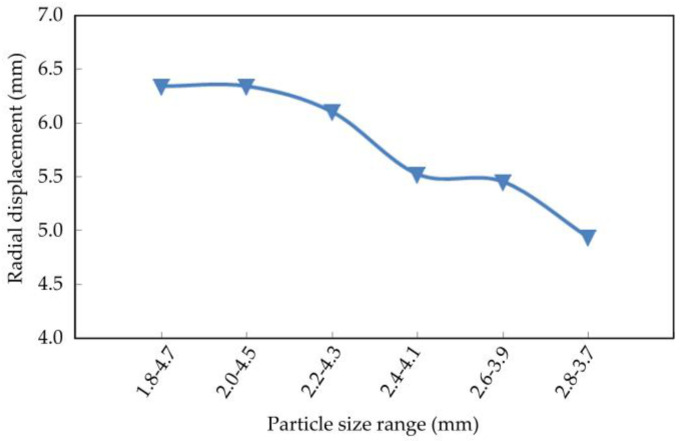
Relationship between radial displacement and particle size.

**Figure 24 materials-16-03465-f024:**
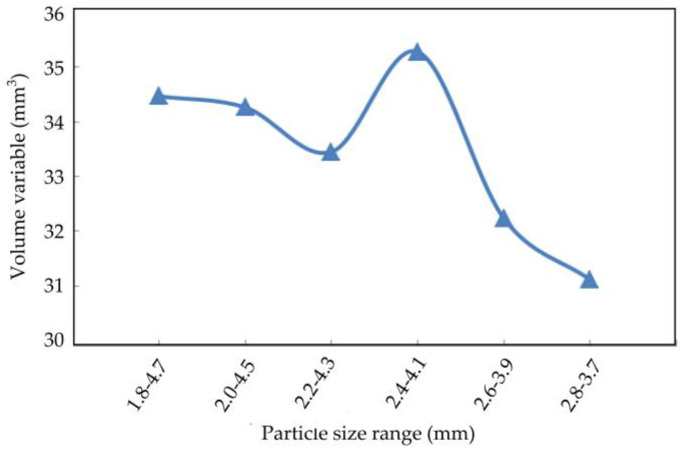
Relationship between volume variables and particle size.

**Table 1 materials-16-03465-t001:** Particle size distribution of samples.

Particle Diameter (mm)	1–2	0.5–1	0.25–0.5	0.1–0.25	0.075–0.1	<0.075
Percentage	6.47%	13.80%	28.73%	17.14%	8.43%	25.43%

**Table 2 materials-16-03465-t002:** Experimental scheme.

Working Condition	Consolidation Pressure (kPa)	Relative Compactness	Vibration Frequency (Hz)	Dynamic Stress Amplitude (kPa)
Working condition 1	100	60	1	70
			1	80
			1	90
			1	145
Working condition 2	100	60	0.5	100
			0.8	100
			1.0	100
			1.2	100
			2	100

**Table 3 materials-16-03465-t003:** Maximum pore water pressure and strain of coral sand with different dynamic stress amplitudes.

Dynamic Stress Amplitudes (kPa)	Pore Water Pressure (kPa)	Strain (%)
70	42.73	0.783
80	45.57	0.954
90	48.01	1.189

**Table 4 materials-16-03465-t004:** Maximum pore water pressure and strain of coral sand with different vibration frequencies.

Vibration Frequency (Hz)	Pore Water Pressure (kPa)	Strain (%)
0.5	49.82	1.112
0.8	55.82	1.121
1.0	58.86	1.189
1.2	61.01	1.371

**Table 5 materials-16-03465-t005:** Macroscopic mechanical parameters of the coral sand sample.

Parameter	Density (g/cm^3^)	Internal Friction Angle (°)	Poisson’s Ratio	Elasticity Modulus (MPa)
Value	1.41	40	0.25	18

**Table 6 materials-16-03465-t006:** Microscopic parameters of the coral sand sample.

Parameter	Normal Stiffness *k*_n_ (N/m)	Tangential Stiffness *k_s_* (N/m)	Friction Coefficient *μ*	Porosity *n*
Value	2 × 10^6^	2 × 10^6^	0.5	0.3

## Data Availability

Data presented in this study are available on request from the corresponding author.

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
