# Peer review of "Laboratory Experiments and Numerical Simulation on Dynamic Response of Island Reclamation Coral Sand under Aircraft Load"

_materials, 2023, doi:10.3390/ma16093465_

Round 1
Reviewer 1 Report
This paper presents the laboratory experiments and numerical simulation on the dynamic response of island reclamation Coral Sand under aircraft load. The reviewer has some minor comments on the results, and the discussion required the authors to rectify in order to improve the merit of the manuscript as follows:
1. Abstract: The abstract is very long with many trivial details. I recommend the authors rewrite the abstract as follows: 1) The gap or the problem in the present state, 2) the methodology used to solve the problem, and 3) the key points and results of the present research.
2. Experimental procedures: why only one value of 100 kPa confining pressure was applied in this study? Why did the authors not explain the failure envelope of the studied materials?
3. Line 243: what does it mean the pore water pressure and strain finally stabilized?
4. Line 250-252: the backup information or reference is required to support this discussion. And the phenomena behind this finding shall be discussed more.
5. Line 300: what is DEM? And Line 307: what is PFC2D?
6. Numerical simulations: the writing on the numerical simulations is not clear. Please consider revising it.
7. What is the difference between micro and microparameters in this study? And how to obtain the macroscopic mechanical parameters of the coral sand?
Reviewer 2 Report
The current study conducts experimental and numerical parametric analysis on coral sands to analyze their dynamic response. To fully explore the dynamic response of coral sand under cyclic dynamic load, a dynamic triaxial compression test taking different dynamic stress amplitudes and vibration frequencies was conducted. Meanwhile, the numerical simulation considered the influence of particle sizes and porosities of coral sand as key variables. The paper shares some interesting results regarding the dynamic response of coral sands. However, the description and discussion of the paper results are limited, making it hard for potential readers to learn some insights from the paper. Therefore, it is recommended to publish the paper only after the authors’ proper adjustments to the paper considering contents and grammatical mistakes.
· The paper should be checked appropriately based on grammatical mistakes such as the use of tense, plural/singular forms of the verb, and the formation of complex sentences to make reading easier and more understandable. For instance, the use of the complex sentence in lines 53-56 and 104-107 should be divided into 2 separate sentences to avoid a grammatical mistake in joining them.
· The literature review, in the introduction section, should contain a critical analysis of the previous work, not just the statements of what other researchers did as in lines 69-70, 75-78, 95-96, and 114-115.
· In subsection 2.3. Aircraft landing load, it is recommended to include a detailed description of how the dynamic response of coral sand 4-6 m below the airport runway during aircraft takeoff and landing.
· During the description of the experimental results, particularly in subsections 2.6.1 and 2.6.2, it is recommended to not just describe the experimental results, but also elaborate on the possible reasons for such results and make some conclusions based on the results.
· Why the maximum and minimum particle sizes of coral sand in numerical analysis are different from that of sand in the experimental study (table 1 and lines 309-310)?
· In section 3 Numerical simulations, the software used in the study should be mentioned.
· The description and discussion of numerical results, subsection 3.5, should include more elaboration on terms that affected the outcome of the study to come to some conclusions.
Round 2
Reviewer 1 Report
The authors have adequately responded to the reviewer's comments and revised the manuscript accordingly. However, a minor comment required to authors to specifically show the microscopic parameters as like the macroscopic parameters in Table 5.
Reviewer 2 Report
The reviewers' comments have been well reflected.
Author Response
Thank you very much.